# In Vitro Analysis of SARS-CoV-2 Spike Protein and Ivermectin Interaction

**DOI:** 10.3390/ijms242216392

**Published:** 2023-11-16

**Authors:** Alejandra García-Aguilar, Rebeca Campi-Caballero, Giovani Visoso-Carvajal, José Rubén García-Sánchez, José Correa-Basurto, Jazmín García-Machorro, Judith Espinosa-Raya

**Affiliations:** 1Laboratorio de Neurofarmacología, Escuela Superior de Medicina del Instituto Politécnico Nacional, Plan de San Luis y Salvador Díaz Mirón s/n, Casco de Santo Tomás, Ciudad de México C.P. 11340, Mexico; alejagarciag8@gmail.com (A.G.-A.); beckycampi@yahoo.com.mx (R.C.-C.); 2Laboratorio de Medicina de la Conservación, Escuela Superior de Medicina del Instituto Politécnico Nacional, Plan de San Luis y Salvador Díaz Mirón s/n, Casco de Santo Tomás, Ciudad de México C.P. 11340, Mexico; carvajalgv@gmail.com; 3Laboratorio de Oncología Molecular y Estrés Oxidativo, Escuela Superior de Medicina del Instituto Politécnico Nacional, Plan de San Luis y Salvador Díaz Mirón s/n, Casco de Santo Tomás, Ciudad de México C.P. 11340, Mexico; jrgarcias@ipn.mx; 4Laboratorio de Diseño y Desarrollo de Nuevos Fármacos e Innovación Biotecnológica, Escuela Superior de Medicina del Instituto Politécnico Nacional, Plan de San Luis y Salvador Díaz Mirón s/n, Casco de Santo Tomás, Ciudad de México C.P. 11340, Mexico; corrjose@gmail.com

**Keywords:** COVID-19, SARS-CoV-2, spike protein, thermolysin, ivermectin, interaction, treatment

## Abstract

The spike (S) protein of SARS-CoV-2 is a molecular target of great interest for developing drug therapies against COVID-19 because S is responsible for the interaction of the virus with the host cell receptor. Currently, there is no outpatient safety treatment for COVID-19 disease. Furthermore, we consider it of worthy importance to evaluate experimentally the possible interaction of drugs (approved by the Food and Drug Administration) and the S, considering some previously in silico and clinical use. Then, the objective of this study was to demonstrate the in vitro interaction of ivermectin with S. The equilibrium dialysis technique with UV–Vis was performed to obtain the affinity and dissociation constants. In addition, the Drug Affinity Responsive Target Stability (DARTS) technique was used to demonstrate the in vitro interaction of S with ivermectin. The results indicate the interaction between ivermectin and the S with an association and dissociation constant of Ka = 1.22 µM^−1^ and Kd = 0.81 µM, respectively. The interaction was demonstrated in ratios of 1:50 pmol and 1:100 pmol (S: ivermectin) by the DARTS technique. The results obtained with these two different techniques demonstrate an interaction between S and ivermectin previously explored in silico, suggesting its clinical uses to stop the viral spread among susceptible human hosts.

## 1. Introduction

The SARS-CoV-2 virus is the etiologic agent of COVID-19 and is responsible for the latest pandemic. Researchers have studied this virus since it first emerged in Wuhan, China [1]. During this time, different health strategies were developed, many of them as an emergency, to reduce the incidence of cases as well as mortality, but also to treat the symptoms of the disease and reduce the damage to the human tissues [2,3,4,5].

Due to the rapid spread of the virus infection around the world and the number of new cases, hospitalizations, and deaths, there was a need to implement strategies to mitigate the entire health problem quickly. In the last couple of years, numerous drugs have been developed, and although they have been approved by the United States Food and Drug Administration (FDA), their long-term adverse effects are unknown (Table 1). The cost of production and distribution of these drugs makes it difficult to acquire them globally [6,7].

In addition to the development of new drugs for the treatment of COVID-19, vaccines against SARS-CoV-2 were developed by several pharmaceutical companies, and their application to the population has been of worthy importance in preventing severe COVID-19, but not as a treatment against infection. In addition, these vaccines do not guarantee their efficacy against the emergence of new variants of SARS-CoV-2 [20]. It is important to mention that several factors influence the efficacy of vaccines, such as demographic characteristics, access to vaccines, immune factors, viral variants, and individual host factors [21].

Currently, only two drugs have been approved by the FDA for emergency use against COVID-19. These medications are Molnupiravir and nirmatrelvir–ritonavir (Paxlovid). Molnupiravir is a prodrug that is metabolized by carboxylesterase in the liver into the cytidine nucleoside N-hydroxycytidine (NHC), then NHC is incorporated into cells where it is phosphorylated to form the active ribonucleoside triphosphate (NHC-TP), which will cause a high rate of errors in the viral genome through viral RNA polymerase, affecting the replication of SARS-CoV-2 [22]. Paxlovid acts at an intracellular level. Nirmatrelvir inhibits the main protease of SARS-CoV-2 (Mpro), thus altering viral replication. Ritonavir acts as a strong inhibitor of cytochrome P 450 (CYP450) specifically CYP3A and prevents the degradation of nirmatrelvir to maintain high levels in the body, allowing it to carry out its mechanism of action [23].

Despite the significant decreases in the number of hospitalized patients or the death rate with the use of these drugs, it is important to emphasize the approval for emergency use against COVID-19, which still requires a lot of research to verify both the safety and the effectiveness of these medications. In the case of Molnupiravir, clinical trials do not report short-term adverse effects, which means that studies must continue to rule out long-term adverse effects. In addition to the fact that its effectiveness has only been demonstrated in patients with mild to high risk, more studies are still needed to determine its effectiveness in patients with mild to moderate COVID-19 [24]. Regarding Paxlovid, there are clinical studies that show a high percentage of safety. However, there are also studies that show that this drug increases cartilage degeneration, causing significant concern about its participation in osteoarthritis, which is why it is recommended, as are Molnupiravir long-term studies on its safety [25]. Then, it is widely justified to repurpose drugs with fewer or without side effects to be used for COVID-19.

In silico studies help to understand the protein–ligand interaction. Therefore, these studies are one of the fastest ways to identify a drug candidate and therapeutic target. Molecular docking is a theoretical simulation method that calculates the binding mode and affinity (free energy values) through force fields, which include bond and non-bond interactions such as electrostatic interactions, hydrogen bond interactions, Van der Waals force interactions, and hydrophobic interactions between molecules (e.g., ligands and receptors) [26]. Different studies have been performed using molecular docking and repositioning potential drugs that could interact with different SARS-CoV-2 proteins, particularly inhibiting the function of spike protein and SARS-CoV-2 entry into host cells to inhibit viral infection [26,27,28].

S protein is the most studied protein from SARS-CoV-2. It is responsible for the interaction of the virus with the host cell receptor by angiotensin-converting enzyme 2 (ACE2), which engages and facilitates viral entry into the cell. It is a highly glycosylated type I transmembrane protein; its molecular weight is 180 to 200 kDa [29,30,31]. The S is divided into two subunits, which have different functions: the S1 subunit is considered to mediate the binding of the virus to the host cell receptor as it contains the receptor binding domain (RBD), and the S2 subunit is responsible for fusing the virus membrane with the host cell membrane [29,30]. The interest in study S ranges from generating vaccines, however, it can be analyzed under drug interactions to find a possible treatment for COVID-19.

Based on previous in silico studies, the drug ivermectin was selected as it has been shown to interact with the S of SARS-CoV-2 through molecular docking [32]. These in silico studies showed that ivermectin could bind to both the S1 and S2 subunits of S. Among the non-covalent interactions between ivermectin and the S1 and S2 subunits were hydrophobic, hydrogen-bonding, and salt-bridging interactions [32]. Also, the ivermectin-S complex yields a binding free energy value of −18 kcal/mol in the presence of the human ACE2 receptor [33].

Ivermectin is an antiparasitic drug used to treat pediculosis, ascariasis, and enterobiasis, among others. It has a high binding affinity to glutamate-regulated chloride ion channels in invertebrate muscles, producing cell hyperpolarization leading to paralysis and death of the parasite [34,35,36,37,38]. Antiviral properties of ivermectin have been documented against different RNA viruses such as human immunodeficiency virus (HIV), influenza, flavivirus, and SARS-CoV-2, as well as DNA viruses such as pseudorabies, polyoma, and adenovirus [39,40]. The antiviral activity of ivermectin is based on its ability to bind and inhibit the transport function of the importin protein (IMPα) from the host, which mediates the nuclear import of several viral proteins. It has been shown that the ORF6 protein of SARS-CoV-2 binds to importin IMPα; however, studies are needed to determine the SARS-CoV-2 proteins that can access the nucleus through the protein importin IMPα [39,41]. The use of ivermectin as a treatment for COVID-19 was approved in some countries, such as Peru and the eastern region of Beni in Bolivia [39,42,43].

In accordance with ClinicalTrials, several reports of clinical studies present results regarding the use of ivermectin alone or in combination with other drugs for the treatment of COVID-19 [44,45,46,47]. In these clinical studies, clinical improvement was observed in patients treated with ivermectin. However, its pharmacological mechanism remains unclear, so it is important to perform more experimental studies to support the use of this drug as a treatment for COVID-19.

Although WHO reported on 4 May 2023 that COVID-19 is no longer considered a public health emergency, the emergence of new virus variants that threaten to reverse the progress made is not ruled out [48]. Although there are new drugs in development for this disease, their long-term effects are unknown, not to mention that the cost of production and distribution makes the acquisition of vaccines and new drugs for COVID-19 challenging to achieve on a global scale.

It is crucial to have an accessible and affordable outpatient pharmacological treatment for COVID-19. The study of the interaction of S with existing drugs approved by the FDA for other diseases allows us to significantly shorten the time and reduce the cost compared with studies for developing new drugs. In addition, these drugs have already been on the market for a long time, facilitating their distribution and access to the population. This work aims to identify whether ivermectin interacts in vitro with the S of SARS-CoV-2 through equilibrium dialysis techniques through UV–Vis spectroscopy and evaluation by the Drug Affinity Responsive Target Stability technique (DARTS).

The dialysis technique allows for determining the binding of a drug to a protein since it is based on the physical separation of the bound ligand from the free one. Dialysis is a process that separates molecules according to their size using semipermeable membranes with pores smaller than those of macromolecules. These pores allow small molecules (i.e., the drug not bound to the protein) to diffuse through the membranes, while the pores of the semipermeable membrane, being smaller, block the passage of larger molecules (i.e., the protein).

The DARTS methodology aims to analyze protein–drug interactions and is based on the principle that when a small molecule binds to a protein, the interaction can stabilize the structure of their target protein such that it becomes protease resistant [49,50].

## 2. Results

### 2.1. Equilibrium Dialysis Assays by UV–Vis

#### 2.1.1. Linearity of the Method

The calibration curve of ivermectin at a wavelength of 240 nm was linear from 10 to 70 µM with a correlation coefficient (r^2^) of 0.99. The equation obtained in this calibration curve was y = 0.01282x − 0.04737. In addition, the calibration curve of the recombinant S1/S2 protein at a wavelength of 240 nm showed linearity at concentrations in the range of 0.23 to 7.5 µM, for which a correlation coefficient (r^2^) of 0.94 was obtained. The equation obtained in this calibration curve was y = 0.07486x − 0.06086.

#### 2.1.2. Protein–Ligand Binding

These data indicate how the absorbance of the bound drug is proportional to the free drug (Table 2, Figure 1). In addition, in a second plot of the reciprocals of bound drug versus free drug (Figure 2), where linearity between bound and free drug was observed, the equation obtained was y = 0.02037x + 0.02486 (r^2^ = 0.90). Additionally, an affinity constant Ka of 1.22 µM^−1^ and a dissociation constant Kd = 0.81 µM were obtained.

### 2.2. Detection of S1/S2 Protein and Ivermectin Interaction by the DARTS Method

The DARTS method is capable of showing the interaction between the recombinant S1/S2 protein and ivermectin. The results show that the recombinant S1/S2 protein interacts with ivermectin. Figure 3 illustrates a representative polyacrylamide gel (the gels were run on three independent occasions) stained with Coomassie blue that shows the protection of S1/S2 protein (132 kDa) in the presence of ivermectin and conditions of proteolyzed protein (Figure 3, lane D and E). In contrast, the protein was degraded in the absence of the drug (Figure 3, lane C). In comparison, no difference was observed in the same samples that underwent mock digestion (Figure 3, lane B). In addition, a protein of 34.6 kDa is expected due to the presence of thermolysin. These data indicated that there is no degradation of the S1/S2 due to its interaction with ivermectin; however, the best protection was observed in the higher ratio of ivermectin interaction (Figure 3, lane E). In addition, the densitometric analysis showed a higher density in the band in lane E compared to the density of the band in lane C, indicating that there is a higher protection of the protein-to-thermolysin activity with the 1:100 ratio of the protein–ligand interaction (Figure 4).

## 3. Discussion

Due to the large number of infections and deaths during the pandemic caused by COVID-19 [51,52], several preclinical, including in silico and clinical trials, were conducted to find a treatment for this disease [53]. Regarding clinical trials, several existing drugs already approved by the FDA for other diseases were prescribed as an emergent treatment due to the lack of an ambulatory and easily accessible treatment for COVID-19. One of these drugs was ivermectin alone or in combination with other drugs, which showed clinical improvement in patients diagnosed with COVID-19 [44,45,54]. These clinical findings show the importance of studying the interaction between ivermectin and the S to achieve progress in establishing an outpatient treatment that helps reduce the symptoms and clinical manifestations caused by COVID-19. Additionally, according to clinical evidence, there are docking studies that report non-bond interactions between S and the drug ivermectin. In addition, these docking studies showed a high affinity constant in terms of the binding free energy of −18 kcal/mol between the receptor binding domain (RBD) of S and the ACE2) receptor [33]. In addition, a binding constant of 5.8 × 10^−8^ is reported, so it is considered that the binding of ivermectin on S may interfere with the binding to the host cell receptor (ACE2) [33]. These experimental assays reported in this work evidence the recognition of ivermectin by S. The data agree with the in silico reports, despite the different affinity values. This is important because, in our study, protein S was used in two versions of equilibrium dialysis: S1/S2 in amino acids 14 to 1213 and S1 in amino acids 20 to 800. Indeed, the association constant is higher when the protein contains the S2 fragment than when it does not (Ka of 1.22 µM^−1^ vs. Ka of 0.094 µM^−1^, respectively). The same is observed with the dissociation constant. There is greater dissociation when the protein contains the S1 fragment than when it contains S1/S2 (Kd of 10.54 µM and Kd of 0.81 µM, respectively), which indicates a higher affinity of ivermectin for S1/S2 of SARS-CoV-2. The affinity of ivermectin to spike protein showed linearity in a double reciprocal plot (Figure 2 and Appendix A). The latter can be compared with an in silico molecular docking study showing that ivermectin binds with higher affinity to the S2 subunit of spike protein by non-covalent interactions [32].

Additionally, the results obtained in this work by the DARTS technique showed that ivermectin interacted in vitro with S1/S2 at a ratio of 1:50 pmol and 1:100 pmol. Therefore, ivermectin protects S1/S2 against its degradation by thermolysin activity, and this is comparable to the interaction between S and thermolysin. It is because, in the absence of the drug, there was degradation of S by thermolysin activity. Despite there being some new drugs proposed for the treatment of COVID-19, more studies are needed to determine the long-term effects that these may present. An example of this is Molnupiravir, since studies report a possible transmission of the virus with mutations due to its use [55]. It is of great importance to study drugs previously approved for other pathologies for which more information is already available on their long-term effects and which are easily distributed and accessible worldwide. To the best of our knowledge, there are no in vitro studies on the direct study of the interaction of S with any FDA-approved drug for other pathologies, so the data obtained in this research can be considered of great importance.

## 4. Materials and Methods

### 4.1. Protein–Ligand Interaction

#### 4.1.1. Spike Protein

Recombinant S, expressed in *E. coli*, was used. The presentation was 1 mg/mL, and it was kept in short-term storage at 4 °C and long-term storage at −80 °C. For the equilibrium dialysis technique, the recombinant proteins Spike S1 and S1/S2 were used. S1 from the manufacturer Virogen (Watertown, MA, USA) includes amino acids from 20 to 800, with a molecular weight of 87.9 kDa. S1/S2, from the manufacturer Invitrogen (Grand Island, NY, USA), contains a region of 14 to 1213 amino acids with a molecular weight of 132 kDa.

#### 4.1.2. Drug and Reagents

Ivermectin (C_95_H_146_O_28_) in powder form with a molecular weight of 875.1 g/mol was purchased from Sigma-Aldrich (St. Louis, MO, USA).

Acetone (spectrophotometric grade) was used as a diluent. It was purchased from Merck (Darmstadt, Germany).

Thermolysin of *Geobacillus stearothermophilus* origin (Sigma-Aldrich, Darmstadt, Germany) was used as a protease in the DARTS technique. A total of 0.5 mg of thermolysin was dissolved in 0.5 mL of distilled H_2_O, which was kept at a temperature of −20 °C.

### 4.2. Equilibrium Dialysis Technique

A total of 0.026 g of ivermectin was dissolved in 30 mL of a mixture of 50%/50% acetone/distilled water. The solution was vortexed and sonicated for 30 min in a Branson 1510 Ultrasonic Cleaner (Branson Ultrasonics Corporation, Richmond, VA, USA). Aliquots of 200 µL of ivermectin were prepared at different concentrations (10 µM, 16 µM, 32 µM, 48 µM, 64 µM, and 70 µM). UV–Vis scanning was performed, for which 2 µL were read in triplicate in a NanoDrop 2000 spectrophotometer (ThermoFisher, Waltham, MA, USA), and the wavelength of 240 nm was determined for the ivermectin standard curve.

For the preparation of the standard curve of the recombinant S1/S2 and S1 spike protein, 20 µL aliquots were prepared at different concentrations (0.23 µM, 0.47 µM, 0.94 µM, 1.8 µM, 3.7 µM, 7.5 µM). After, UV–Vis scanning was performed, for which 2 µL were read in triplicate in a NanoDrop 2000 spectrophotometer. The wavelength was selected at 240 nm.

The equilibrium dialysis technique was performed based on the Ligand Binding Module manual [56] and the articles by Correa-Basurto A et al. and García et al. [57,58]. However, some modifications were made briefly. Equilibrium dialysis was carried out with different concentrations of ivermectin (10 µM, 16 µM, 32 µM, 48 µM, 64 µM, and 70 µM); the drug was obtained from the 1 mM stock solution and was volumetrically buffered with phosphate-buffered saline (PBS, 7.2 pH). The drug concentrations were placed in an external solution at 80 mL. In contrast, a Slide-A-Lyzer™ G2 gamma-ray dialysis cassette (Thermo Scientific, Rockford, IL, USA), with a membrane that prevents the passage of molecules larger than 20 kDa, was used for the retention of S1/S2 protein. The membrane was hydrated for 24 h before use; a volume of 360 µL with a constant concentration (0.94 µM) of recombinant S1/S2 protein, diluted in buffer (PBS) with a pH of 7.2, was placed inside the membrane (Appendix A). The experiment was kept at 4 °C and kept in a steering rack (Thomas Scientific, Swedesboro, NJ, USA). Samples of 200 µL of the external solution were taken every 20 min for 3 h and 20 min. At the end of the dialysis, the sample contained inside the membrane was removed, and all the samples obtained were conserved at 4 °C. UV–Vis performed a triplicate reading of 2 µL of the samples obtained in a NanoDrop 2000 spectrophotometer at a wavelength of 240 nm. After analysis, the samples were stored at −80 °C.

#### Determination of Dissociation and Association Constants

For the quantification of the ligand bound, the absorbance of the bound ligand (∆A) was calculated using the following equation: ∆A = A − A0, the final absorbance obtained in the protein–ligand complex (A) was subtracted from the initial absorbance S (A0), and the absorbances were obtained at 240 nm. Subsequently, the absorbances of the bound drug were determined using the general equation of a straight line (y = mx + b) from the standard curve of ivermectin at 240 nm (y = 0.01282x − 0.04737). The drug bound to the complex was subtracted from the total drug to calculate the free drug. A double reciprocal plot was made with the reciprocal of the bound ligand absorbance against the free ligand reciprocal (Table 2, Figure 1). In addition, affinity (Ka) and dissociation (Kd) constants were calculated based on the Ligand Binding Module manual and the article by Nafisi et al. [56,59]. The following equations were used: Ka = intercept/slope and Kd = 1/Ka, equivalent to Kd = slope/intercept. The values for S1/S2 were obtained from the equation y = 0.02037x + 0.02486, which was obtained from the graph of the reciprocal of the bound ligand against the reciprocal of the free ligand (Table 2 and Figure 2). The values for S1 were obtained from the Appendix A, graph of the reciprocal of the bound ligand against the reciprocal of the free ligand (Appendix A) show the equation y = 0.1011x + 0.009587.

### 4.3. DARTS Technique with S1/S2 Protein and Ivermectin

For the performance of the DARTS method, five samples were prepared in 0.5 mL tubes. Sample 1: 10 µg (75.6 pmol) of recombinant S1/S2 protein. Sample 2 (control): 10 µg of S1/S2 protein plus 1.5 µL of 10× buffer reaction (500 mM Tris (pH = 8), 500 mM NaCl, 100 mM CaCl_2_) plus 1.8 µL of H_2_O distilled plus 1.6 µL of ivermectin, equivalent to a ratio of 1:50 pmol (the drug was taken from a stock solution of 1 mg of ivermectin diluted in 500 µL of acetone). Sample 3 (protein plus thermolysin) was prepared to test the thermolysin activity in the absence of the drug: 10 µg of S1/S2 protein plus 1.5 µL of 10× buffer reaction plus 3.2 µL of distilled H_2_O. Sample 4: identical amounts as the control sample were placed to determine the interaction at a ratio of 1:50 pmol. Sample 5: 10 µg of S1/S2 protein plus 1.5 µL 10× buffer reaction plus 0.2 µL of distilled H_2_O and 3.3 µL of ivermectin were added to determine the binding of the protein with the ivermectin at a 1:100 pmol ratio. The amount of ivermectin was taken from the stock solution. For the interaction with ivermectin, samples 2 (control), 4 (1:50 pmol), and 5 (1:100 pmol) were incubated for 12 h at a temperature of 37 °C.

Subsequently, 0.3 µL of thermolysin (a ratio of 0.3 μg of thermolysin for every 10 μg of protein) was added to samples 3, 4, and 5 and allowed to react for 10 min at 37 °C. The reaction was stopped by adding 5 µL of EDTA at 0.5 M concentration in samples 2, 3, 4, and 5, which were placed on ice. Samples were loaded on gels prepared with 4% acrylamide gel concentrador (0.533 mL acrylamide, 1 mL buffer, 2.466 mL H_2_O, 40 µL 10% ammonium persulfate, and 4 µL TEMED), while the separator gel was 8% acrylamide (1.6 mL acrylamide, 2 mL buffer, 0.8 mL 80% glycerol, 1.6 mL H_2_O, 30 µL 10% ammonium persulfate, and 3 µL TEMED), and electrophoresis was performed for 90 min at 100 V in a BIO-RAD Mini-PROTEAN system (Bio-Rad Laboratories, Inc., Hercules, CA, USA). The gel was stained with Coomassie Brilliant Blue for 25 min. Finally, it was destained for 24 h with a destaining solution of methanol (50%), acetic acid (10%), and distilled water (40%) to visualize the bands in the gel (Figure 3). The DARTS method was performed according to Lomenick et al. [60,61]; however, modifications were made to the interaction time and concentrations.

Densitometry analysis of the bands present in the D and E reels was performed to determine the density of the amount of protein present in each protein–ligand interaction; the thermolysin band was used as a loading control. ImageJ software (ImageJ 1.53t, Wayne Rasband, National Institutes of Health, Bethesda, MD, USA) was used for the analysis.

## 5. Conclusions

The data, obtained through two different techniques, suggested an interaction between S and ivermectin. These data are of worthy importance since there is very little information on completed clinical studies using ivermectin as a treatment against COVID-19, and we did not find in vitro studies concerning the interaction of ivermectin with the S. The results obtained are novel and may contribute to the selection of drugs in future in vivo studies and advance determining an ambulatory treatment against COVID-19.

## Figures and Tables

**Figure 1 ijms-24-16392-f001:**
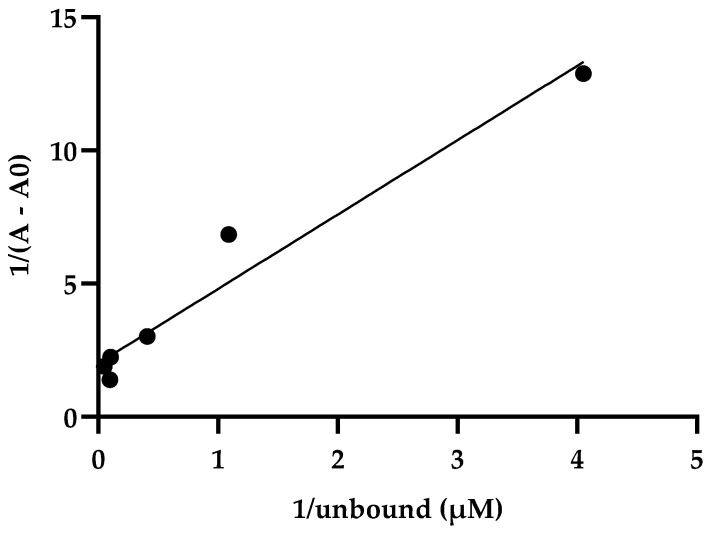
The double reciprocal plot of 1/(A − A0) versus 1/unbound. Linearity is observed between the reciprocal of the absorbance of the bound ligand (A − A0) versus the free ligand reciprocal at different concentrations. Each dot indicates the intersection of the values of column 1 (reciprocal absorbance, ligand bound) with columns 3 (reciprocal ligand unbound) of Table 2.

**Figure 2 ijms-24-16392-f002:**
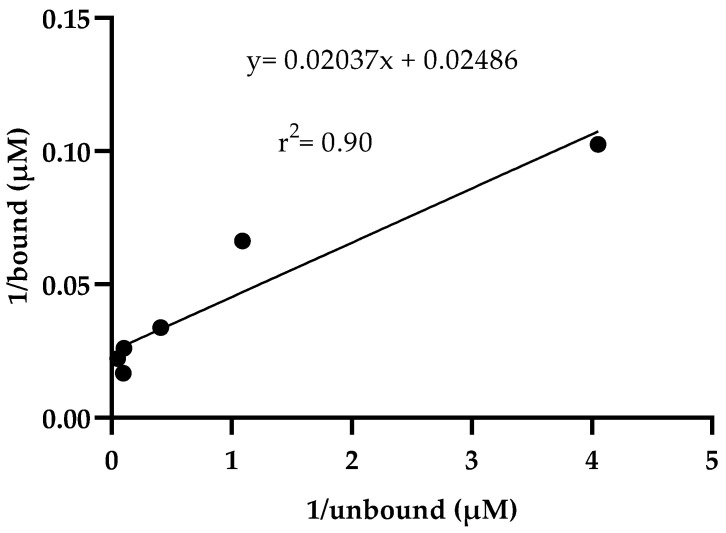
The double reciprocal plot of 1/bound ligand versus 1/unbound ligand. Linearity is observed between the reciprocals of bound and free ligand, Ka = 1.22 µM^−1^ and Kd = 0.81 µM, according to the equation: y = 0.02037x + 0.02486 and (r^2^ = 0.90). Each dot indicates the intersection of the values of column 2 (reciprocal ligand bound) with columns 3 (reciprocal ligand unbound) of Table 2.

**Figure 3 ijms-24-16392-f003:**
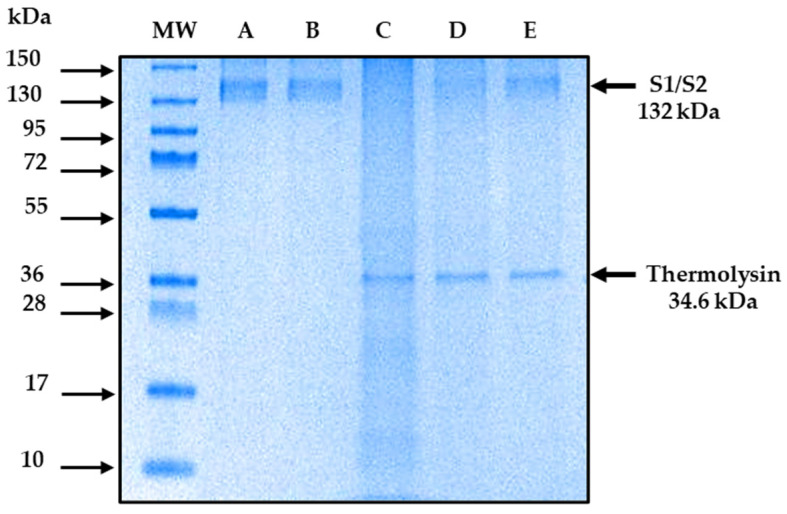
DARTS uses recombinant S1/S2 protein and ivermectin. Coomassie blue-stained SDS-polyacrylamide (8%) gel. Lane A, recombinant S1/S2 protein; lane B, S1/S2 protein plus ivermectin (mock digestion); lane C, S1/S2 protein plus thermolysin; lane D, S1/S2 protein plus ivermectin (1:50 ratio) plus thermolysin; lane E = S1/S2 protein plus ivermectin (1:100 ratio) plus thermolysin. The arrows indicate the proteins; notice that S1/S2 protein was degraded when ivermectin was absent and in the presence of thermolysin (lane C). The molecular weight (MW) and molecular masses of the markers (kDa) are given on the sides of the gel.

**Figure 4 ijms-24-16392-f004:**
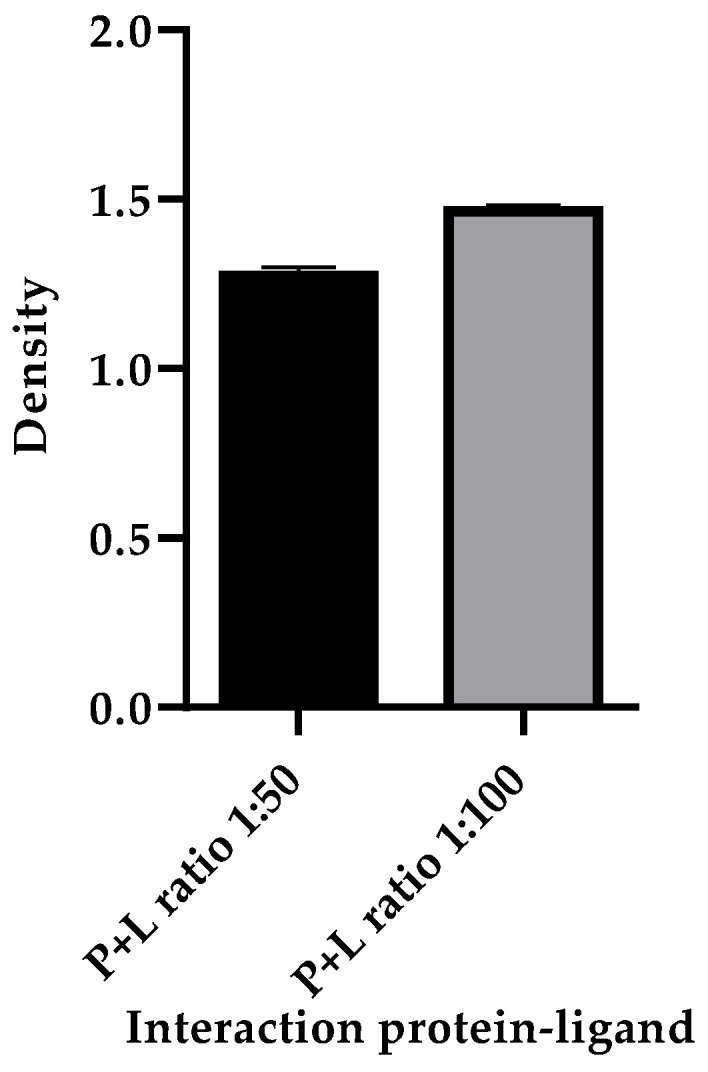
Densitometric analysis. Data represent mean densities ± SD from three independent experiments. A lower density was observed in the 1:50 ratio protein–ligand interaction band compared to the 1:100 ratio interaction band, indicating that the protection of protein S against thermolysin activity is higher in the 1:100 ratio protein–ligand interaction. Thermolysin is resistant to these experimental conditions, so it was used as a loading control.

**Table 1 ijms-24-16392-t001:** Specific FDA-approved treatments for COVID-19.

Drug	Mechanism of Action	Route of Administration	Recommendations	References
Remdesivir	Inhibition of RNA polymerase, blocking viral replication	I.V.	For hospital use For adult and pediatric patients older than 12 years and weighing > 40 kg	[8,9,10]
Tocilizumab	Recombinant humanized monoclonal IgG1 anti-IL6 receptor antibody	I.V.	For hospital use For adult and pediatric patients 2 years and older under treatment with systemic corticosteroids and supplemental oxygen	[11,12]
Nirmatrelvir/ritonavir	Nirmatrelvir: Inhibits * 3CLpro of SARS-CoV-2 Ritonavir: ** CYP3A inhibitor	V.O.	For use in patients with mild to moderate COVID-19, in adults and pediatric patients over 12 years of age and weighing > 40 kg and testing positive for SARS-CoV-2	[13,14,15,16]
Molnupiravir	Introduces errors in the genetic code of SARS-CoV-2, preventing its replication	V.O.	For adult patients testing positive for SARS-CoV-2 and at risk of developing severe COVID-19	[17,18,19]

I.V.: Intravenous; V.O.: Oral route; IgG1: Immunoglobulin G; IL6: Interleukin 6; * 3CLpro: Protease involved in viral replication; ** CYP3A: Enzyme involved in the metabolism of Nirmatrelvir; inhibiting it increases the concentration of Nirmatrelvir in the body for a longer period of time.

**Table 2 ijms-24-16392-t002:** Ligand-bound and free ivermectin at different concentrations with recombinant S1/S2 protein.

Reciprocal Absorbance Ligand Bound 1/(A − A0)	Reciprocal ^1^ Ligand Bound (µM) 1/(Bound)	Reciprocal ^1^ Ligand Unbound (µM) 1/(Unbound)
1.3882	0.0166	0.0988
1.8891	0.0222	0.0525
2.2438	0.0260	0.1048
3.0150	0.0338	0.4108
6.8493	0.0662	1.0910
12.8755	0.1025	4.0526

**^1^** Values obtained for the reciprocals of the bound and unbound ligand in the recombinant S1/S2 protein with ivermectin using the UV–Vis equilibrium dialysis technique.

## Data Availability

Data is contained within the article and Appendix A.

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
