# Peer review of "In Vitro Analysis of SARS-CoV-2 Spike Protein and Ivermectin Interaction"

_ijms, 2023, doi:10.3390/ijms242216392_

Round 1
Reviewer 1 Report
Comments and Suggestions for Authors
Overall, this manuscript demonstrates in vitro analysis of SARS-CoV-2 spike protein and ivermectin interaction. However, after reviewing, this work is not good enough to be published in IJMS due to the following concerns.
1. The working volume is not sufficient for a publication. In this paper, only one experiment was carried out. Besides, the significance of this work is not clear based on the paper.
2. In the introduction part, the authors mentioned “Currently, there is no treatment for COVID19 in ambulatory patients endorsed by the World Health Organization (WHO) and the FDA”. However, Pfizer’s paxlovid is a good example for COVID treatment, and it was not introduced in the paper.
3. The authors determined association and dissociation constants, which cannot be both µM. The unit of K11 should be wrong.
4. The experimental design is too simple. Since there are some mutations of spike protein, will there be an effect on affinity? Since Protein S can bind to ACE2 protein, will the small molecule added affect the binding? There should be some competitive assay.
5. The authors used one in vitro method to determine the binding affinity, however, there are a lot of different technologies for drug-protein interaction study. The authors didn’t compare with other methods, and no information found in the introduction or discussion part.
Comments on the Quality of English LanguageModerate editing of English language required.
Reviewer 2 Report
Comments and Suggestions for Authors
In this manuscript, Alejandra García-Aguilar et al. demonstrate the in vitro interaction of ivermectin with SARS-CoV-2 spike protein, with both Equilibrium Dialysis and Drug Affinity Responsive Target Stability (DARTS).
The overall English level of the manuscript is good, with a few typos to address. For this purpose, as well as for the correction of minor issues, please refer to the attached revised version of the manuscript, in which those issues are highlighted in yellow and addressed with a comment.
The introduction is well written but, on the other hand, the results section severely lacks.
Starting from the performed experiments, even though they are well consolidated techniques, the authors do not even outline them, and just give references to the readers. I find that even a short description of the principles of both Equilibrium Dialysis and DARTS is necessary to give context to the selected research design, to strengthen the reader’s understanding. This is especially true for the Equilibrium Dialysis where, in the results sections, graphs are presented without any explanation on how they were obtained and of what information can be gained from each one of them. To strengthen this part, the authors must first explain the principle of the technique and then give more details about each of the presented data in the results section itself.
Regarding the DARTS experiment, I also have some concerns:
1) How many times was the experiment repeated? The authors state (line 181) that the gel provided in Figure 3 is a representative one, so they must clearly state how many experiments were performed;
2) Even tough spike protein protection from proteolysis by ivermectin is appreciable in the shown gel, I think the authors could include (in Figure 3) a densitometric analysis of the DARTS experiments, quantifying the spike protein band signal and using the thermolysin one as a loading normalizer;
3) I am not sure the gel shown in Figure 3 has an actual 8% polyacrylamide amount. Please check this matter and eventually clarify it.
Furthermore, and most importantly, I have some methodologic concerns that I would like the authors to address for this manuscript to be considered for publication.
First, different proteins are used for different experiments: the recombinant spike S1 protein (amino acids 20-800) for the Equilibrium Dialysis and the recombinant spike S1/S2 protein (amino acids 14-1213) for DARTS. Is there a reason for this choice? I cannot find it in the manuscript, and I think the authors must clarify this point for two reasons:
1) the authors cite the work from Choudhury et al., in which molecular docking shows that ivermectin could bind to both the SARS-CoV-2 spike protein S1 and S2 subunits but with a higher strength to S2. S1 comprises the amino acids 14-685, whereas S2 the amino acids 686-1273. Thus, with ivermectin showing a higher affinity for S2, why did the authors decide to perform Equilibrium Dialysis with a recombinant spike S1 protein (amino acids 20-800) comprising only a little part of S2? This must be clarified;
2) Wanting to compare the affinity data (Equilibrium Dialysis) with the evidence of the in vitro interaction obtained through DARTS, using two different proteins which bear different portions of S1 and S2 is confusing. Thus, if no major issue hampers it (which should be consistently explained in the manuscript), the author should use the same protein for both experiments and, if possible, select for this purpose the one used for DARTS, as it is more representative of the actual full-length spike protein. If this is not possible, they should include another DARTS experiment, alongside the one already shown, using the recombinant spike S1 protein they used for Equilibrium Dialysis. I think such a simple add-on experiment could greatly benefit the paper strength.
Moving forward, even though I think that this work is worthy of showing the direct interaction of ivermectin with SARS-CoV-2 protein Spike, I also think that the experimental section is a bit shrunken and would benefit few additional experiments. As a suggestion, for a better comparison with the Equilibrium Dialysis data, the authors could perform thermal stability assays on the recombinant S1/S2 spike protein, first incubating the protein with (or without) ivermectin at several temperatures, to then select the optimal one to perform an isothermal titration with the molecule. Western Blotting could be used as a read out (as an example, please refer to Jafari et al., The cellular thermal shift assay for evaluating drug target interactions in cells, DOI: 10.1038/nprot.2014.138).
Regarding the Materials and Methods:
1) at line 237 I do not understand where this recombinant protein was used, given that afterwards the authors state that for equilibrium dialysis they used the manufacturer Virogen recombinant spike S1 protein, and for DARTS the recombinant spike S1/S2 protein, from Invitrogen. This is just very confusing and must be clarified;
2) At line 261 I think that E-07 µM is clearly a mistake, but it appears several times in the manuscript (e.g., line 274). Please correct it.
Thus, considering all the above reported concerns, I think submission of this work must be re-considered after major revisions.

The overall English level of the manuscript is good, with a few typos to address.
Author Response
Please see the attcahment

Round 2
Reviewer 1 Report
Comments and Suggestions for Authors
Although the previous manuscript showed several major problems, the revised/re-submitted version is greatly improved. It has already addressed most of the concerns in the previous review comments. The description on the data matches the experimental design. In addition, more details have been added to the result section, which provides more comprehensive discussion.
There are still some minor changes that need to be done (e.g., in discussion part, binding constant 5.8 E-8, please revise to keep the format identical; also, the concentration units in the second paragraph of section 4.2 seem not right). Please carefully check all the abbreviations, spellings, and units to make sure they are correct.
Overall, the revised manuscript is suitable for publication in IJMS after minor revision.
Comments on the Quality of English LanguageMinor editing of English language required.
Reviewer 2 Report
Comments and Suggestions for Authors
I gratefully thank the Authors for addressing my concerns. Still, few typos remain to correct (e.g., line 262, uM must be added; line 268, y must be corrected), but I think the paper can be accepted for publication after a final careful proofreading of the text.
Comments on the Quality of English LanguageA moderate English language editing is needed.